# A methodological framework for deriving the German food-based dietary guidelines 2024: Food groups, nutrient goals, and objective functions

Anne Carolin Schäfer[1,2]*, Heiner Boeing[3], Rozenn Gazan[4], Johanna Conrad[1], Kurt Gedrich[5], Christina Breidenassel[1], Hans Hauner[6], Anja Kroke[7], Jakob Linseisen[8], Stefan Lorkowski[9,10], Ute Nöthlings[2], Margrit Richter[1], Lukas Schwingshackl[11], Florent Vieux[4], Bernhard Watzl[12]

1 German Nutrition Society, Bonn, Germany, 2 Department of Nutrition and Food Sciences, Nutritional Epidemiology, University of Bonn, Bonn, Germany, 3 Department of Epidemiology (closed), German Institute of Human Nutrition Potsdam-Rehbruecke, Nuthetal, Germany, 4 MS-Nutrition, Marseille, France, 5 Research Group Public Health Nutrition, ZIEL—Institute for Food & Health, Technical University of Munich, Freising, Germany, 6 Institute of Nutritional Medicine, Else Kröner Fresenius Center for Nutritional Medicine, School of Medicine and Health, Technical University of Munich, Munich, Germany, 7 Department of Nutritional, Food and Consumer Sciences, University of Applied Sciences, Fulda, Germany, 8 Epidemiology, University of Augsburg, University Hospital Augsburg, Augsburg, Germany, 9 Institute of Nutritional Sciences, Friedrich Schiller University Jena, Jena, Germany, 10 Competence Cluster for Nutrition and Cardiovascular Health (nutriCARD), Halle-Jena-Leipzig, Germany, 11 Institute for Evidence in Medicine, Medical Center—University of Freiburg, Faculty of Medicine, University of Freiburg, Freiburg, Germany, 12 Department of Physiology and Biochemistry of Nutrition, Max Rubner-Institut, Karlsruhe, Germany

* corresponding_author@dge.de

## Abstract

### Background

For a growing number of food-based dietary guidelines (FBDGs), diet optimization is the tool of choice to account for the complex demands of healthy and sustainable diets. However, decisions about such optimization models' parameters are rarely reported nor systematically studied.

### Objectives

The objectives were to develop a framework for (i) the formulation of decision variables based on a hierarchical food classification system; (ii) the mathematical form of the objective function; and (iii) approaches to incorporate nutrient goals.

### Methods

To answer objective (i), food groups from FoodEx2 levels 3-7 were applied as decision variables in a model using acceptability constraints (5th and 95th percentile for food intakes of German adults ($n = 10,419$)) and minimizing the deviation from the average observed dietary intakes. Building upon, to answer objectives (ii) and (iii), twelve models were run using decision variables from FoodEx2 level 3 ($n = 255$), applying either a linear or

**Data availability statement:** We share the position that data being used to produce the article should be in principle publicly available. However, in our case, we did not generate data in the context of this article that were not shown in the tables and supplements. The main data sources were from third parties that can easily be contacted via the information given in the text and references. The main data sources that have been used are listed below: Software: • DGE, parts of own developed program solutions can be made available upon request (contact at https://www.dge.de/) • MS-Nutrition has provided the basic programming and should be consulted if interested in the program code (contact at https://ms-nutrition.com/en/) Data: • Nutrient database: The data on nutrients used for obtaining the results presented in the manuscript are available for the Bundeslebensmittelschlüssel (BLS) from the Max RubnerInstitut (contact: https://blsdb.de/contact) and for LEBTAB from the University of Bonn (https://www.epi.uni-bonn.de/forschung/donald-studie, contact via noethlings@unibonn.de). • Food intake data: The data on food intake underlying the results presented in the study are available from the European Food Safety Authority (EFSA) which has published national consumption data at https://www.efsa.europa.eu/en/data-report/food-consumption-data and can be accessed via a public access per data and documents request (for details, see https://www.efsa.europa.eu/sites/default/files/2023-08/pad-guidance-for-applicants.pdf) All relevant data on changes applied to the original databases (e.g. whole grain classification) are mentioned within the paper and its Supporting Information files.

**Funding:** This research was partly funded by the German Federal Ministry of Food and Agriculture in the form of payment of salary to JC, MR, CB and ACS. The funder had no role in the decisions about data collection, analyses, interpretation of data, in the writing of the report nor in the decision to submit the article for publication.

**Competing interests:** The authors have declared that no competing interests exist.

squared and a relative or absolute way to deviate from observed dietary intakes, and three different lists of nutrient goals (allNUT-DRV, incorporating all nutrient goals; modNUT-DRV excluding nutrients with limited data quality; modNUT-AR using average requirements where applicable instead of recommended intakes).

## Results

FoodEx2 food groups proved suitable as diet optimization decision variables. Regarding deviation, the largest differences were between the four different objective function types, e.g., in the linear-relative modNUT-DRV model, 46 food groups of the observed diet were changed to reach the model's goal, in linear-absolute 78 food groups, squared-relative 167, and squared-absolute 248. The nutrient goals were fulfilled in all models, but the number of binding nutrient constraints was highest in the linear-relative models (e.g. allNUT-DRV: 11 vs. 7 in linear-absolute).

## Conclusion

Considering the various possibilities to operationalize dietary aspects in an optimization model, this study offers valuable contributions to a framework for developing FBDGs via diet optimization.

## 1. Introduction

Food-based dietary guidelines (FBDGs) are typically developed to advice the public by which food intake dietary reference values (DRVs) can be met [1], and more recently, to reduce their risk of diet-related non-communicable diseases [2,3]. Nowadays, more advanced FBDGs integrate additional dimensions such as ecological sustainability [4,5], which increases the complexity of defining optimal intakes of different food groups [6,7].

The most holistic approach to manage the complex task of evaluating multiple dimensions of a sustainable diet is diet optimization, also referred to as diet modeling or linear/quadratic (squared) programming [7–10]. It aims to find the optimal quantitative combination of food groups (the decision variables), that may have both conflicting and/or complementary features, and that fulfills a set of constraints while minimizing or maximizing an objective function. Such a tool has been successfully applied in the context of FBDG development, for example in Australia, France, and the Netherlands [7,11–14].

However, each parameter in the optimization model can be represented in different (mathematical) options [15]. The food groups considered for FBDGs, and thus as decision variables for mathematical optimization, are often determined on an ad hoc basis [2,6,16,17]. The European Food Safety Authority (EFSA) has set up a detailed food classification system, FoodEx2, which is used to analyze and report survey data for European countries [18–21], what makes its use as decision variables favorable. However, it is unclear how such hierarchically organized food coding systems perform in optimization models. The objective function defines indicator(s), which should be minimized or maximized. In the literature using diet optimization, the most frequent indicator is the observed diet from which the deviation is minimized and thereby maximizes cultural acceptability of the model's solution [22–24]. This fulfills the requirement that FBDGs should take the habitual diet of the population into account [2]. Constraints define the solution space in which the model can operate. When designing nutritionally adequate diets using diet optimization, they are usually set to fulfill

nutrient recommendations using nationally available DRVs [12,25] and to adhere to acceptability limits, where food group quantities are limited to realistic consumption levels observed in the population, e.g., the 95[th] percentile of consumption [8,12].

Although mathematical optimization has become an established method in nutritional sciences, hardly any systematic methodological analyses have been reported. With the goal of updating and expanding the German FBDG methodology, we undertook thorough analyses of these methodological choices. The present paper aimed to describe the use of FoodEx2 as decision variables for optimization models, the implication of the objective function's mathematical form taking the observed dietary intake from the latest German nutrition survey as example, and the impact of different choices of nutrient goals based on DRVs on optimization results.

## 2. Methods

### 2.1. Decision variables

The decision variables are the variables that will be optimized. Optimization models for FBDGs typically focus on observed dietary intakes for a list of food groups as decision variables. For Germany, the food classification of EFSA, FoodEx2, provides internationally comparable food intake data and was therefore selected as the basis of our decision variables (version MTX 12.1, Exposure hierarchy [26]). Following a parent–child hierarchy, the FoodEx2 food groups are ordered into seven levels, with level 1 being the most aggregated (e.g., "Fruit and fruit products") and the lower levels being more detailed (e.g., "Pome fruits" on level 3) (Fig 1). Level 7 is the level at which intake data from surveys such as the most recent German National Nutrition Survey II [20,27] had been initially coded before being further aggregated for communication and comparisons. FoodEx2 facets, which further describe attributes of a food group, e.g., processing information, were not considered in this study.

However, FoodEx2 does not allow a clear distinction between whole grain and other grain products without the use of FoodEx2 facets. To work with well-defined food groups for whole grains, the food groups that belong to level 1 "Grains and grain-based products" were relabeled to obtain the food groups "whole grain" and "refined grain" as displayed in S1 Table. The relabeling was conducted by duplicating the initial "Grains and grain-based products" food group to create two different categories ("Whole grains" and "Refined grains") that still

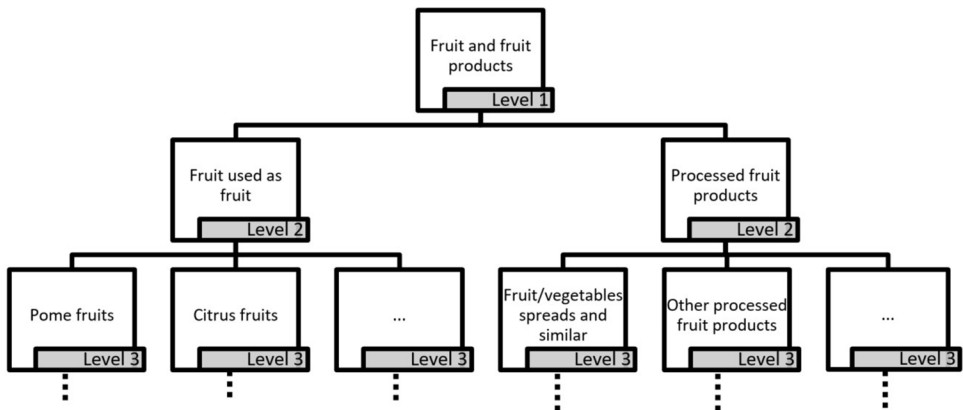

**Fig 1. Example of the hierarchical structure of FoodEx2 showing food groups from level 1 "Fruit and fruit products".**

adhered to the parent–child hierarchy. All initial food groups at level 7 were categorized as whole grain or refined grain and then aggregated until level 1.

Processed meat is also not clearly defined in the hierarchy of FoodEx2, as several level 2 food groups of the parent group "Meat and meat products" are a mix of processed meat products and unprocessed meat. Here, a new parent food group called "Processed meat" was established and the respective FoodEx2 food groups were assigned to this group. To differentiate between poultry and red meat for the food group "Mammals and birds meat" (level 3, food code A0EYH), the food code was duplicated, reassigned to either red meat or poultry, and the mean intake split between the two new variables. The percentiles used as acceptability constraints (chapter 2.2) were kept as observed.

The FoodEx2 level 1 groups "Other ingredients" and "Food products for young population" were excluded for their irrelevancy and low intakes by adults. After these adjustments, the number of food groups, and therefore decision variables, was 927 at level 7, which were aggregated into 926 at level 6, 857 at level 5, 593 at level 4, 255 at level 3, and 83 at level 2.

**2.1.1. Reporting the whole diet.** Considering that FoodEx2 level 2 already has 83 different food groups, the optimization results would be too detailed for establishing and communicating FBDGs. Thus, we decided that the reporting of results from our optimization models should be condensed into a defined list of food groups. This list was agreed upon by the working group of the German Nutrition Society responsible for the development of the scientific basis for the German FBDG (S2 Table). This list included the following groups: water, coffee and tea, vegetables, fruits, fruit and vegetable juices, legumes, nuts and seeds, potatoes, grain (products) and wholegrain (products) thereof, milk and dairy products, eggs, fish and seafood, poultry, red meat, processed meat, vegetable oils, and spreadable fats.

These 18 FBDG food groups were matched to their most aggregated respective FoodEx2 food groups. Eight of the 18 food groups matched FoodEx2 level 1 food groups, six matched level 2 food groups, and four matched level 3 food groups (Table 1).

Many FBDGs aim to limit the intake of so-called discretionary foods [2], such as sweets and sugar-sweetened beverages. These food groups contribute to total energy intake and the intake of nutrients that should be limited, such as free sugars and saturated fatty acids. These heterogeneous and unfavorable food groups were reported as the summary of their energy share in addition to the 18 FBDG food groups. Discretionary food groups and their respective FoodEx2 food groups are displayed in Table 2.

**2.1.2. Selection of the appropriate FoodEx2 level for optimization.** To determine which level of FoodEx2 should be selected as the source of decision variables for the optimization of the German FBDG, we concluded that the level should provide sufficient details to create a specific result for each food group reported in the FBDG. For example, if the FBDG should report the optimal consumption quantity of vegetable oils, as distinguished from other fats at level 3 (Table 1), level 3 of FoodEx2 would need to be selected as the source of decision variables. We found that FoodEx2 level 3 (255 decision variables) provided a list of food groups with sufficient detail for reporting in the German FBDG (Table 1). The list of decision variables at level 3 is shown in S3 Table.

**2.1.3. Linking food groups to consumption and nutrient composition data.** In the EFSA Comprehensive European Food Consumption Database, the FoodEx2 food codes are already linked to dietary intake data from various national surveys [19,20]. For our purpose, we retrieved data for adults (18–65 years) in Germany. The German intake data is based on two non-consecutive 24-hour dietary recalls from 10,419 men and women from the German National Nutrition Survey II (NVS II) (2005-2007) [29]. The data was weighted as described in Heuer et al. for age, sex, residential area, and other socioeconomic factors to represent the adult German population [30]. For each FoodEx2 food group and level, the mean intake and

**Table 1. Food-based dietary guidelines (FBDG) food groups and respective FoodEx2 food group names and codes.**

| FBDG food group | FoodEx2 food group name | FoodEx2 code | FoodEx2 level |
|---|---|---|---|
| *Drinking water* | Drinking water | A03DK | 2 |
| *Coffee and tea* | Ingredients for coffee, cocoa, tea, and herbal infusions | A03GH | 2 |
| *Vegetables* | Vegetables and vegetable products | A00FJ | 1 |
| *Fruit* | Fruit and fruit products | A01BS | 1 |
| *Fruit and vegetable juices* | Fruit and vegetable juices and nectars (including concentrates) | A039K | 1 |
| *Vegetable fats and oils* | Vegetable fats and oils, edible | A036N | 3 |
| *Legumes* | Legumes | A04RG | 2 |
| *Nuts and seeds* | Nuts, oilseeds and oilfruits | A04RH | 2 |
| *Potatoes* | Potatoes and similar | A0DPP | 3 |
| *Whole grains** | Grains and grain-based products | A000J_WG* | 1 |
| *Refined grains*** | Grains and grain-based products | A000J_RG** | 1 |
| *Eggs and egg products* | Eggs and egg products | A031E | 1 |
| *Fish and seafood* | Fish, seafood, -amphibians, reptiles and invertebrates | A026T | 1 |
| *Milk and dairy products* | Milk and dairy products | A02LR | 1 |
| *Poultry* | Birds meat | A0EYG | 3 |
| *Red meat* | Mammals meat | A0EYF | 3 |
| *Processed meat**** | Processed meat products | A01QR_P*** | 2 |
| *Spreadable fats* | Fat emulsions and blended fats | A039B | 2 |

*All grain food groups that contain "wholemeal," "bran," "brown," or "oat" in their name (based on [28]) and individual case decisions as noted in S1 Table.

**All grain food groups except food groups that contain "wholemeal," "bran," "brown," or "oat" in their name (based on [28]) and individual case decisions as noted in S1 Table.

***Generated from the food groups "Processed whole meat products," "Sausages," "Meat specialities," and "Canned-tinned meat".

The FoodEx2 level 1 food groups "Products for young population" and "Other ingredients" were excluded.

**Table 2. Discretionary food groups according to FoodEx2 food group names and codes.**

| Food group | FoodEx2 food group name | FoodEx2 code | FoodEx2 level |
|---|---|---|---|
| *Seasoning and sauces* | Seasoning, sauces and condiments | A042N | 1 |
| *Composite dishes* | Composite dishes | A03VA | 1 |
| *Sugar-sweetened beverages* | Water-based beverages | A04PY | 2 |
| *Alcoholic beverages* | Alcoholic beverages | A03LZ | 1 |
| *Sweets* | Sugar and similar, confectionary and water-based sweet desserts | A032F | 1 |
| *Others* | Products for non-standard diets, food imitations and food supplements; FoodEx2 food groups not able to be matched to other groups | A03RQ | 1 |

distribution percentiles in g/d were calculated, either for those individuals who consume the food group (consumers only) or among all individuals.

Information on nutrient and energy content was obtained from the German Nutrient Database (Bundeslebensmittelschlüssel (BLS) version 3.02 [31]). The BLS uses a different food group classification system than FoodEx2. To match the intake data with the national nutrient database, a matching of 1,288 foods from the BLS to FoodEx2 food groups, mainly on level 7, was applied. As the BLS lacks data for free sugars, this information was completed using mean values from 80 generic food categories of the LEBTAB database [32] that were matched to BLS food groups.

The nutrient values for all food groups (decision variables) were calculated following the parent–child hierarchy of FoodEx2 (Fig 1). For a specific food group, the nutritional content

was either the nutritional content of a BLS food if a direct matching existed, or an average nutritional content of its related food groups according to the FoodEx2 hierarchy that had a matching with the BLS database, weighted by intake data, as described by Gazan et al. [33].

## 2.2. Acceptability constraints

Acceptability constraints frame the solution space of the optimization model to ensure that optimized food intakes remain within the range observed in the target population. They were defined as follows: the 5th (minimum constraint) and 95th percentiles (maximum constraint) of the observed intake for all individuals for each food group at level 1 (see S4 Table) and, additionally, for each food group on level 3, the 5th (minimum constraint) among all individuals and 95th percentiles (maximum constraint) for consumers only.

For "Composite dishes" and "Seasoning and sauces," which are part of the discretionary food groups, we made exceptions from the aforementioned rules. These food groups are not clearly described (e.g., potato-based dishes), but supply various nutrients within one decision variable, posing a very attractive group for the linear models which strive to achieve the best solution by changing the least decision variables. Therefore, the optimized intake for these food groups could not exceed the corresponding observed mean intakes on level 3. Because the energy goal of the optimized diet is lower than the observed diet, the observed intakes used as upper acceptability constraints were matched to the energy goal of the optimized diet.

## 2.3. Objective function and nutrient goals

### 2.3.1. Different implementations of deviation from the observed diet.
Four different mathematical implementations for minimizing deviation from observed dietary intakes in the objective function were investigated: linear as a percentage from the observed diet (linear-relative), linear in absolute quantities (linear-absolute), squared differences of the percentage from the observed diet (squared-relative), and squared differences of the absolute quantities (squared-absolute).

The mathematical formulas were as follows:

$$Linear-relative\ Dev_{lr}\ =\ \sum_{i=1}^{n}\frac{\|(x_i^{opt}-x_i^{obs})\|}{x_i^{obs}}$$

$$Linear-absolute\ Dev_{la}\ =\ \sum_{i=1}^{n}\|(x_i^{opt}-x_i^{obs})\|$$

$$Squared-relative\ Dev_{sr}=\sum_{i=1}^{n}\left(\frac{x_i^{opt}-x_i^{obs}}{x_i^{obs}}\right)^2$$

$$Squared-absolute\ Dev_{sa}\ =\ \sum_{i=1}^{n}\left(x_i^{opt}-x_i^{obs}\right)^2$$

where *Dev* is the deviation from the observed diet composed of *n* food groups, with $x_i^{obs}$ the observed quantity of food group *i* and $x_i^{opt}$ the optimized quantity of the same food group.

### 2.3.2. Three approaches to incorporate nutrient goals.
The nutrient goals were based on the dietary reference values (DRVs) for Germany [34] and complemented by EFSA's tolerable upper intake levels for nutrients [35]. DRVs are classified as either recommended intakes (RI), estimated values, or guiding values. Nutrients with an RI cover the nutrient requirements of

97.5% of the population [34]. The average requirement (AR) for these nutrients covers the requirements of 50% of the population. Three different lists of nutrient goals were defined by setting lower bounds, upper bounds, or specific target values. If not differently specified, reference values for adults (18–65 years, normal weight with moderate physical activity level (PAL) 1.4) were taken and subsequently weighted according to the proportion of sex and distribution of age groups in the German population (S5 Table).

The first list applied all nutrient goals (DRVs und EFSA upper intake levels) and where available the RI (allNUT-DRV). Selenium, chromium, and molybdenum were not included because there is no data in the nutrient database used (BLS version 3.02). Vitamin D was not included because the requirement is not to be covered by diet, but provided by the endogenous synthesis through exposure to sunlight or by supplements.

The second list (modNUT-DRV) built upon allNUT-DRV, but was modified in three different ways: (i) iodine, fluoride, copper, and manganese were not applied due to limited data quality in the nutrient database. Their contents vary greatly depending on their fortification in foodstuffs or animal feed or are not assessed at all; the contribution of other sources, e.g., fluoride intake from toothpaste, is unknown; and/or the bioavailability fluctuates (manganese) or is homeostatically regulated (copper). As the quantities tend to be underestimated rather than overestimated, the upper bounds were nevertheless used. (ii) Regarding total fat, an upper bound was applied (40% of energy intake) [36,37]. It was assumed that lower and upper bounds of fatty acids (saturated fatty acids (SFA), monounsaturated fatty acids (MUFA), and polyunsaturated fatty acids (PUFA)) would ensure adequate quality and intake of fat [34,37]. (iii) For iron, the DRV for premenopausal women was used for the entire population instead of a weighted mean for both sexes because this is the only DRV with a higher value for women and a higher risk for insufficient supply. For similar reasons, the upper limit for alcohol was taken from the women's DRVs.

The third list of nutrient goals applied goals for the same nutrients as modNUT-DRV, except that here the AR was used instead of the RI (modNUT-AR). The remaining DRVs (11 estimated values, 5 guiding values, and 4 other recommended intakes) remained the same. A detailed overview of all lists is provided in Table 4 in the Results section.

**Table 3. Results of the deviation indicators between the observed diet and optimized models for the allNUT-DRV, modNUT-DRV, and modNUT-AR models in all four objective function types.**

| Deviation indicator | allNUT-DRV | | | | modNUT-DRV | | | | modNUT-AR | | | |
|---|---|---|---|---|---|---|---|---|---|---|---|---|
| | Linear | | Squared | | Linear | | Squared | | Linear | | Squared | |
| | Rel. | Abs. | Rel. | Abs. | Rel. | Abs. | Rel. | Abs. | Rel. | Abs. | Rel. | Abs. |
| Sum of absolute changes in food groups (g/day) | 5,158 | 493 | 5,468 | 729 | 2,784 | 464 | 3,413 | 673 | 3,191 | 427 | 2,003 | 576 |
| Sum of relative changes in food groups (%) expressed as average per variable (n = 255) | 35 | 1,234 | 51 | 20,763 | 23 | 965 | 35 | 20,421 | 16 | 3785 | 24 | 15,275 |
| No. of food groups (n = 255) that increased in the optimized diet | 14 | 8 | 90 | 108 | 11 | 7 | 81 | 116 | 8 | 6 | 70 | 114 |
| No. of food groups (n = 255) that decreased in the optimized diet | 16 | 11 | 55 | 34 | 15 | 13 | 68 | 31 | 16 | 13 | 74 | 37 |
| No. of food groups (n = 255) that disappeared in the optimized diet | 19 | 54 | 25 | 106 | 20 | 58 | 18 | 101 | 13 | 55 | 11 | 96 |
| No. of food groups (n = 255) that did not change in the optimized diet | 206 | 182 | 85 | 7 | 209 | 177 | 88 | 7 | 218 | 181 | 100 | 8 |
| No. of level 1 groups (n = 18) that reached maximum acceptability constraints | 2 | 0 | 3 | 0 | 2 | 0 | 1 | 0 | 2 | 0 | 0 | 0 |

**Table 4. Nutrient goals for all analyses and nutrient contents for the observed diet and optimized diet in the linear-relative model for all nutrient goals (allNUT-DRV), the modified list of nutrient goals (modNUT-DRV), and the modified list of nutrient goals using the average requirement (AR) (modNUT-AR).**

| Nutrient (unit) | Type | Obs. | allNUT-DRV | | | modNUT-DRV | | | modNUT-AR | | |
| --- | --- | --- | --- | --- | --- | --- | --- | --- | --- | --- | --- |
| | | | Lower bound | Upper bound | Opt. | Lower bound | Upper bound | Opt. | Lower bound | Upper bound | Opt. |
| Fat (% of energy) | GV | 39 | – | – | 30 | – | 40** | 29 | – | 40** | 29 |
| Saturated fatty acids (% of energy) | n.a. | 17 | – | 10 | 10 | – | 10 | 10 | – | 10 | 10 |
| Monounsaturated fatty acids (% of energy) | n.a. | 13 | 10 | – | 10 | 10 | – | 10 | 10 | – | 10 |
| Polyunsaturated fatty acids (% of energy) | n.a. | 6 | 7 | 10 | 8 | 7 | 10 | 7 | 7 | 10 | 7 |
| Linoleic acid (% of energy) | RI | 4.8 | 2.5 | – | 6.7 | 2.5 | – | 6 | 2 | – | 6 |
| Linolenic acid (% of energy) | EST | 0.7 | 0.5 | – | 0.8 | 0.5 | – | 0.8 | 0.5 | – | 0.7 |
| EPA plus DHA (mg/d) | EST | 289 | 250 | – | 674 | 250 | – | 274 | 250 | – | 334 |
| Cholesterol (mg/d) | GV | 318 | – | 300 | 300 | – | 300 | 300 | – | 300 | 218 |
| Protein (g/d) | RI | 78 | 52 | – | 76 | 52 | – | 82 | 43 | – | 67 |
| Carbohydrates (% of energy) | n.a. | 43 | – | – | 52 | – | – | 52 | – | – | 55 |
| Free sugars (% of energy) | n.a. | 15 | – | 10** | 10 | – | 10** | 10 | – | 10** | 10 |
| Fiber (g/d) | GV | 18 | 30 | – | 30 | 30 | – | 30 | 30 | – | 30 |
| Alcohol (ethanol) (g/d) | GV | 12 | – | 15 | 6 | – | 10 | 6 | – | 10 | 6 |
| Vitamin A (µg RAE/d) | RI | 1,534 | 776 | – | 1,144 | 776 | – | 1,025 | 550 | – | 1,196 |
| Vitamin D (µg/d) | EST | 2.6 | – | 100* | 5 | – | 100* | 2.4 | – | 100* | 1.7 |
| Vitamin E (mg/d) | EST | 13 | 13 | 300* | 20 | 13 | 300* | 16 | 13 | 300* | 20 |
| Vitamin K (as phylloquinone) (µg/d) | EST | 83.1 | 68 | – | 105 | 68 | – | 81.7 | 68 | – | 104.4 |
| Thiamin (mg/d) | RI | 1.7 | 1.1 | – | 1.8 | 1.1 | – | 2 | 0.9 | – | 1.8 |
| Riboflavin (mg/d) | RI | 2 | 1.2 | – | 1.8 | 1.2 | – | 1.9 | 1 | – | 1.5 |
| Niacin (mg/d) | RI | 37 | 13 | – | 36 | 13 | – | 38 | 11 | – | 34 |
| Pantothenic acid (mg/d) | EST | 5.9 | 5 | – | 6.4 | 5 | – | 6.4 | 5 | – | 5.9 |
| Vitamin B6 (mg/d) | RI | 2 | 1.5 | 25* | 2.3 | 1.5 | 25* | 2.3 | 1.3 | 25* | 2.5 |
| Biotin (µg/d) | EST | 70 | 40 | – | 86 | 40 | – | 75 | 40 | – | 78 |
| Folate (µg/d) | RI | 278 | 300 | 1,000* | 377 | 300 | 1,000* | 349 | 220 | 1,000* | 331 |
| Cobalamin (Vitamin B12) (µg/d) | EST | 6.5 | 4 | – | 5.5 | 4 | – | 5.2 | 4 | – | 4.5 |
| Vitamin C (mg/d) | RI | 128 | 103 | – | 210 | 103 | – | 136 | 84 | – | 237 |
| Sodium (mg/d) | EST | 2,507 | 1,500 | 2,400** | 2,400 | 1,500 | 2,400** | 2,400 | 1,500 | 2,400** | 2,215 |
| Chloride (mg/d) | EST | 3,873 | 2,300 | – | 4,539 | 2,300 | – | 4,036 | 2,300 | | 3,620 |
| Potassium (mg/d) | EST | 3,191 | 4,000 | – | 4,000 | 4,000 | – | 4,000 | 4,000 | – | 4,000 |
| Calcium (mg/d) | RI | 1,015 | 1,000 | 2,500* | 1,000 | 1,000 | 2,500* | 1,000 | 741 | 2,500* | 741 |
| Phosphorus (mg/d) | RI | 1,375 | 700 | – | 1,312 | 700 | – | 1,445 | 580 | – | 1,093 |
| Magnesium (mg/d) | EST | 360 | 325 | – | 426 | 325 | – | 490 | 325 | – | 382 |
| Iron (mg/d) | RI | 11.3 | 12 | – | 13.1 | 15 | – | 15 | 11 | – | 12.5 |
| Iodine (µg/d) | RI | 106 | 193 | 600* | 193 | – | 600* | 111.3 | – | 600* | 87.9 |
| Fluoride (mg/d) | GV | 1.1 | 3.5 | 7* | 3.5 | – | 7* | 1.9 | – | 7* | 1.5 |
| Zinc (middle phytate intake) (mg/d) | RI | 11.3 | 11 | 25* | 11 | 11 | 25* | 12.3 | 9.2 | 25* | 9.9 |
| Copper (mg/d) | EST | 1.7 | 1 | 5* | 2 | – | 5* | 2.3 | – | 5* | 1.9 |
| Manganese (mg/d) | EST | 5.1 | 2 | – | 9.9 | – | – | 7.1 | – | – | 5.5 |
| Water (mL/d) | GV | 2,946 | 2,156 | – | 5,803 | 2,156 | – | 4,623 | 2,156 | – | 4,185 |

Obs. = observed diet, Opt. = optimized diet, RI = recommended intake, EST = estimated value for an adequate intake, GV = guiding value, EPA = eicosapentaenoic acid, DHA = docosahexaenoic acid, RAE = retinol activity equivalent, AR = average requirement.

DRVs from [34], unless differently indicated;

*European Food Safety Authority Dietary Reference Value/Upper Intake Level;

**German Nutrition Society (Deutsche Gesellschaft für Ernährung) recommendation [42–44].

The only target values applied were for energy, a guiding value with a target value of 2,029 kcal per day for every model, and fat with a target of 30% of energy per day in the allNUT-DRV model. In all analyses, vitamin D was not considered due to its mainly endogenous synthesis.

**2.3.3. Mathematical implementation of nutrient goals in the optimization model.** In addition to minimizing deviation from observed dietary intakes, the objective function of models considering nutrient goals had an additional component that only became active when a nutrient goal could not be fulfilled:

$$Dev_k^- = \begin{cases} \dfrac{DRV_k^{min} - Intake_k^{opt}}{DRV_k^{min}} & if\ Intake_k^{opt} \leq DRV_k^{min} \\ \\ 0 & if\ Intake_k^{opt} > DRV_k^{min} \end{cases}$$

$$Dev_l^+ = \begin{cases} \dfrac{Intake_l^{opt} - DRV_l^{max}}{DRV_l^{max}} & if\ Intake_l^{opt} \geq DRV_l^{max} \\ \\ 0 & if\ Intake_l^{opt} < DRV_l^{max} \end{cases}$$

where for nutrients $k$ with a minimum nutrient goal, only the inadequate intake (negative deviations) $Dev_k^-$ is minimized (a), and for nutrients $l$ having a maximum nutrient goal, only excess (positive deviations) $Dev_l^+$ is minimized (b). For nutrients with a target goal (e.g., a target of 2,000 kcal/d), the sum of negative and positive deviations is minimized. This allows deviation from nutrient goals if the model would otherwise find no feasible solution, which helps to identify the source of infeasibility. It should be noted that in the case of the squared function $Dev_l^+$ and $Dev_k^-$ are squared.

**2.3.4. Complete objective function and technical aspects.** The objective function used for investigations in the present work aims to minimize $F_j$, which is the sum of the total deviation from each specific objective:

$$Min\ F_j = Dev\ + W^N \left[ \sum_{k=nutrients \atop min}^{k=1} Dev_k^- + \sum_{l=nutrients \atop max}^{l=1} Dev_l^+ + \sum_{m=nutrients \atop target}^{m=1} \left( Dev_m^- + Dev_m^+ \right) \right]$$

A high weight $W^N$ was assigned to penalize the objective function value if the solution deviated from nutrient goals.

The diet optimization models were developed using R version 4.1.3 [38], with an R package specifically designed for this project using the ROI (R Optimization Infrastructure) package version 0.3-3 [39], the solver lpsolve [40] for linear optimization, and quadprog version 1.5.8 [41] for quadratic/squared optimization. The data of the descriptors were stored in an SQL database and edited using MySQL Workbench version 8.0.

## 2.4. Analysis

To study the impact of the use of the hierarchical food code FoodEx2 as decision variables, a simple optimization model was used that only included the minimization from the observed diet in the objective function, acceptability constraints, and no nutrient constraints. To study the impact of the mathematical type of objective function ($Dev_{lr,la,sr,sa}$) and varying nutrient goals, 12 scenarios of the complete optimization model were run using decision variables from FoodEx2 level 3 with all four objective function types, acceptability constraints, and the allNUT-DRV, modNUT-DRV, and modNUT-AR lists for nutrient goals. To study how the mathematical implementation type used for objective function affected the results, the following indicators were used to measure deviation from the observed diet:

- The absolute sum of changes in food groups (grams/day):

$$Sum \ of \ absolute \ changes = \sum_{i=1}^{n} \| \left( x_i^{opt} - x_i^{obs} \right) \|$$

- The sum of relative changes in food groups(%):

$$Sum \ of \ relative \ changes = \sum_{i=1}^{n} \| \left( x_i^{opt} - x_i^{obs} \right) \| / x_i^{obs} \times 100$$

- The number of food groups that increased in the optimized diet

- The number of food groups that decreased in the optimized diet

- The number of food groups that disappeared in the optimized diet

- The number of food groups that did not change in the optimized diet

- The number of level 1 groups that were binding acceptability constraints.

Another criterion for the comparison between the models was the number of binding nutrient constraints defined as those constraints where the nutrient quantity in the optimized diet was equal to the imposed upper or lower bound [9].

## 3. Results

### 3.1. Use of a hierarchical food code as decision variables

Intuitively, a food code constructed according to a parent–child hierarchy will generate similar solutions for the decision variables independent of the applied level. We validated this principle by running optimization models with decision variable sources ranging from level 7 to level 2 (S6 Table). In this context, a low level of aggregation in FoodEx2 meant a higher number of decision variables was used for optimization; this was counterbalanced by uniform reporting of results, often on a highly aggregated level (see the Methods section). The optimization program had no difficulties in processing the 927 decision variables of level 7 or other levels and generated equal amounts for each reported food group regardless of the applied level. These findings suggest that the selected R programs represent a powerful tool for detailed food consumption data, such as those categorized in FoodEx2.

However, whether food group level affects the results when further parameters are applied (such as nutrient constraints) was not investigated, but was rated according to the experience with optimization models as minor. The relative independence of level may be explained by the fact that due to the clear parent–child relationship from level to level, each level contains the same information but is subdivided differently.

### 3.2. Different types of objective functions

With the outlined metrics, we compared the optimized diet for four different optimization functions and three nutrient goal lists with the observed diet (Table 3). In the allNUT-DRV models, the absolute and relative sum of changes in food groups was higher than those in the modNUT-DRV and modNUT-AR models. However, the largest differences in the deviation indicators were between the four different objective function types. As defined in the objective function, the sum of absolute quantitative changes was higher in the relative models than in the absolute models: for example, a total of 5,158 g of food for the linear-relative allNUT-DRV model vs. 493 g for the linear-absolute allNUT-DRV model. Consequently, the relative changes were considerably higher in the absolute models than in the relative ones (e.g., an average of 965% per variable for the linear-absolute modNUT-DRV model vs. 23% for the linear-relative modNUT-DRV model).

The absolute models for allNUT-DRV, modNUT-DRV, and modNUT-AR reached none of the acceptability limits on FoodEx2 level 1 (limits shown in S4 Table). All relative models except for the modNUT-AR squared-relative model reached one to three acceptability limits. In the squared-relative allNUT-DRV model, for example, these were maximum limits for "Vegetables," "Drinking water," and "Coffee and tea." The linear-relative modNUT-DRV model, which used high quantities of beverages as energy-independent variables to increase nutrient content, did not reach the acceptability constraints for these groups, but did for "Potatoes" and "Legumes and nuts."

The linear models generated results that left the largest number of decision variables unchanged (relative ($Dev_{lr}$): 206–218; absolute ($Dev_{la}$): 177–182 out of 255 decision variables), whereas the squared-relative models left less than half of the variables (85–100) unchanged and the squared-absolute models left only 7–8 food groups unchanged. The latter also showed the highest number of food groups disappearing from the diet (96–106 food groups).

### 3.3. Nutrient goals

Table 4 shows the nutrient contents of the optimized diets for the linear-relative objective function. The optimized nutrient contents across the other objective function forms were similar and are shown in S6 Table. All nutrient goals were met in all models. In the allNUT-DRV, modNUT-DRV, and modNUT-AR models, the constraints for the six nutrients SFA and free sugars (upper limits), and MUFA, fiber, potassium, and calcium (lower limits) were binding, meaning that they reached exactly their upper or lower bound. In the allNUT-DRV model (15 upper and 33 lower bounds), cholesterol, sodium, iodine, fluoride, and zinc were also binding constraints. In the modNUT-DRV approach, there were four additional binding nutrient constraints (cholesterol, sodium, PUFA, and iron) from a total of 16 upper bounds and 28 lower bounds. The modNUT-AR model had the same number of nutrient goals, but only PUFA as another binding constraint. After excluding nutrient goals due to limited data quality in the modNUT-DRV and modNUT-AR models, the DRV was no longer reached for iodine and fluoride, and even the AR [34] was not reached for iodine in the modNUT-AR model. For copper and manganese, the DRV was met despite exclusion of their intake goals.

## 4. Discussion

In the present study, we showed different ways food groups as decision variables, the objective function, and nutrient goals of optimization models used for deriving FBDGs can be defined, and explored their implications on optimization results. We also investigated the implications of using a hierarchical food code as the source of decision variables for FBDG development, showing that selecting an appropriate food code, based on the desired level of detail for the FBDGs, is essential for the model. Further, we revealed the impact of using different mathematical forms of the objective function on dietary changes and provided reasons for making decisions about objective function types in the future. Lastly, we clarified which nutrient goals are suitable for use in the model. These analyses paved the way for constructing an optimization model that forms the basis of Germany's FBDG 2024.

This study provides methodological investigations of specific aspects of diet optimization which we were faced with in the process to tailor the final model for the German FBDG. We could not find similar methodological comparisons of different settings when studying other country's FBDGs. This could be due to the fact that in most other FBDGs based on optimization results such as the FBDGs in France, Australia, and for Malawian children, iterative approaches to introduce new parameters to the models were used instead of comparisons of

different parameters settings [11,12,14]. We found the mentioning of sensitivity analyses for different objective functions in terms of minimizing deviation from the current diet or from certain nutrients were carried out, but results were not reported [14], or the food groups were studied in preliminary optimization testing, but again no methodological details or insights of this exercise were given [11]. For the Dutch FBDG, only the final selected settings were reported without comparisons with alternative scenarios [13]. The described optimization model could handle several hundred food groups (ranging from 255 at level 3 to 927 at level 7) without interfering with the internal parent–child hierarchy of the FoodEx2 food code (S6 Table). Subsequently, deciding on which level of FoodEx2 the optimization should be run is vital. We showed that the same optimized food intake quantities, independent of which food group level was initially selected, were obtained in an optimization model minimizing the distance from the observed intake data only and without nutritional constraints (S6 Table), demonstrating the hierarchical consistency of the food code data.

Previous studies on optimization have mostly used national food codes, selected codes of intake survey data, or a predefined food list to generate a list of decision variables, determined considering characteristics such as nutrient composition or dietary habits [11,12,17,45,46]. The food classification system in this study supports a harmonized approach of data extrapolation, expansion, and use; it matches various food data sources, striving to harmonize nutritional research not only in European countries but also worldwide [47,48]. There are several reasons why a well-accepted and internationally used hierarchical system like FoodEx2 is useful for diet optimization: (i) it provides clearly defined food groups at each level; (ii) it provides representative and regularly updated food intake data, which can be easily incorporated and compared; (iii) gaps in the data can be calculated using dietary intake as weighting factors (see Methods section, [33]); and (iv) solutions can be evaluated by researchers of different disciplines working with such food code data [47]. To the best of our knowledge, a hierarchical system like FoodEx2 has neither been applied to diet optimization nor systematically studied as decision variables. One challenge in the application of FoodEx2 with regards to FBDGs is that the exposure-rooted classification does not always fit with a nutritional–physiological perspective (e.g., no clear distinction between whole grain and refined grain products); therefore, adjustments, like those described in the methods section, may be needed.

Another question addressed the selection of an appropriate aggregation level of food groups as decision variables. In this analysis, using FoodEx2 level 3 as the source of decision variables was adequate to calculate optimized intakes with the required detail to quantify each of the 18 food groups reported as output of the optimization model and basis of the FBDG, while also allowing for the highest possible level of aggregation. In the type of parent–child hierarchy seen in FoodEx2, the aggregated characteristics of food groups are similar across levels but have different degrees of detail. Therefore, the largest possible aggregation of food groups is required to minimize the chance of single food groups biasing the representativeness of the food group. As an example, increasing the intake of the nutrient-dense food group liver could lead to a significantly smaller amount of total meat than would have been the case without this variable. The preference for more aggregated food groups that better represent a general group over variables with specific properties has been described previously [25]. If a different degree of detail is required, e.g., if differences in citrus fruit intake needed to be addressed, choosing a different food group level would be necessary. Therefore, the food groups about which statements are to be made define the ideal FoodEx2 level of decision variables for optimization.

Further use of optimization results, e.g., for the development of FBDGs, usually target generic food groups that reflect the public's understanding and are suited for easily understandable communication [16]. These food groups are in general much broader than the

single estimates for food groups generated by the optimization model. Thus, we decided on a uniform reporting schema of the optimization results independent of the level and details of the model itself. We decided to also report the optimal intake of discretionary foods. In our case, they were essential to remain consistent with the FoodEx2 classification system in our optimization model. Some nutrient goals, especially upper limits for SFA, alcohol, and free sugars, particularly focus on discretionary food groups. These food groups are often not considered in diet modeling for FBDG derivation [13,49]. However, there are also examples of the integration of discretionary foods in diet modelling for the Australian FBDG [11] and food guides from the Netherlands and Belgium [13,50].

Above, we mentioned the tendency of the model to select specific foods that favor the fulfillment of optimization functions such as nutritional constraints due to their characteristics. Thus, it is important to set appropriate constraints for each food group at all levels. In our case, similar to earlier studies, the 5th and 95th percentiles of observed food intakes were selected as minimum and maximum constraints to keep optimized quantities within acceptable intake ranges [51]. To date, no other study has provided rational criteria to assess the acceptability of these limits [8,52,53]. Borgonjen et al. [46] found that the application of the 10th and 90th percentiles increased the number of nutrients that were difficult to fulfill and recommended the use of the 5th and 95th percentiles, while also stating that a narrower range of percentiles would more closely represent average food patterns, making the recommendations easier to adopt. A narrow range of acceptability constraints can also limit the flexibility of the model to derive healthy diets, once indicators for diet–health relations are incorporated: using the 90th percentile for all individuals as acceptability constraints, the model could, for example, only use a maximum of 202 g/d for "Vegetables," whereas the 95th percentile cutoff is 263 g/d (applied in this study), which is closer to an optimal vegetable intake regarding health [28]. Therefore, for our model we selected the 5th and 95th percentiles as a compromise between limiting unrealistic results and providing flexibility to fulfill the model's goals and, in the future, health-oriented goals.

Regarding the mathematical form of the objective function, while linear approaches seem to be the most common choice for most optimization problems [54] and in the French, Australian, and Malawian FBDG for children [11,12,14], the reasoning for this choice of objective function is rarely found [8]. For the Dutch FBDG, it was claimed that quadratic functions are superior in terms of cultural acceptability because smaller changes in more food groups would be easier to accept [13]. In the present study, we compared both relative and absolute linear and quadratic objective functions. However, our modeling could only partly confirm these theoretical considerations. The number of unchanged food groups was the highest in the linear-relative model, demonstrating that this model produces larger changes in fewer food groups, in accordance with the literature [8,13]. As expected, the quadratic approaches led to smaller changes in more food groups, but the sums of these changes, when considered both absolutely and relatively, were similar to or greater than the sums of changes from the linear models. Furthermore, particularly in the squared-absolute models, the quadratic functions led to several food groups being excluded from the diet on the decision variable level, which mostly affected food groups with dietary intakes < 1 g/d. Previously, where a quadratic approach was used, the smallest observed intakes were a priori rounded down to 0 g/d [55].

When deviations from observed diets in absolute quantities were compared with relative quantities, the absolute models were expected to make smaller changes. The absolute model does not consider the scales of food group quantities, whereas the relative model gives each decision variable a similar weight and therefore tends to favor food groups consumed in high quantities. The results showed that these assumptions held true and indeed the deviation for the absolute models was smaller. However, a relative approach can assure that popular foods

are used to meet the model's requirements: in the linear-absolute allNUT-DRV model, "Tea leaves derivatives and tea ingredients," a rarely consumed (observed intake < 0.1 g/d) and dry concentrated component, was used to fulfill the goal for fluoride. By contrast, large amounts of coffee beverages (observed intake ~ 700 g/d) were used in the linear-relative model. Therefore, a lower deviation from the observed diet does not necessarily mean that the optimized diet is more practicable.

We observed that each mathematical type of objective function had specific properties that directed the model to find the optimal solutions. Which model performs best is therefore subject to the stated hypothesis and context of the analyses. In our case, we wanted to use the model results to guide the development of the new FBDGs for Germany. Therefore, we hypothesized that larger changes to often-used food groups would be the best strategy to communicate the new guidelines, rather than focusing on small changes to many food groups or eliminating some rarely used food groups. The type of function that met these criteria best was the linear-relative function, which was selected as our approach.

Considering the observed dietary intake via the objective function and as acceptability constraints fulfilled the EFSA framework's requirement to consider this aspect for the derivation of FBDGs [2] and increases the acceptability of the optimization results. However, the term acceptability should be handled carefully: the acceptability of a diet in a population is not only represented by the smallest possible deviation from the mean observed diet, but can also be displayed in other characteristics such as financial cost [53]. Without application in real-life studies, as done previously [56], it is impossible to define which objective function is most acceptable. Further, the observed diet represents no single individual's dietary pattern, but an average of all survey participants. Hence, even the starting point may not be acceptable to many people: the most acceptable option for an individual will always be a personal decision [9,25].

The fulfillment of specific nutrient goals strongly drives optimization results and hence the scientific basis for FBDGs. Therefore, a thorough investigation of the selection and impact of different nutrient goals was undertaken. All nutrient goals could be fulfilled in the allNUT-DRV, modNUT-DRV, and modNUT-AR models. However, with more and higher nutrient goals (allNUT-DRV> modNUT-DRV> modNUT-AR), a greater number of binding nutrients constraints appeared. Binding nutrients are most difficult for the model to fulfill and strongly drive the optimization results. Most often, they appear when the observed diet failed to meet a nutrient goal. On the other hand, due to the mutual influence of the model's parameters with rising complexity also apparently uncritical nutrients or constraints can cause challenges for the model: E.g., the upper bound for SFAs forced the model to change fat-providing foods to fulfil both the nutrient goal constraints for MUFAs and SFAs. Thereby, the lower bound for MUFAs became a binding constraint, although this goal was initially met in the observed diet. Another optimization study found that constraints for SFAs were difficult to fulfill with subsequent changes in fatty acid profile [57]; similar observations were found for vitamin E and sodium combined with potassium DRVs in the US [58–60].

Comparing the three different approaches to incorporate nutrient goals regarding deviation from the observed diet, we found the biggest difference between the modified lists and the first list using all nutrient goals. The latter led to solutions with higher deviations from the observed diet. These findings were expected because the first list is more restrictive. This is in line with other optimization studies that tend to exclude or adapt single nutrients that drive strong deviations [25,61]; modifying the list of nutrient goals ensures optimized diets do not increase rarely consumed food groups for nutrients that are not well-represented in the data anyways [9]. Furthermore, there often is a trade-off between incorporating a more restrictive

nutrient constraint and acceptability. Less acceptable results can also stem from limited data quality for certain nutrients (see Chapter 2.4), which in this study was resolved by excluding the affected nutrient goals in the modNUT-DRV and modNUT-AR lists.

The allNUT-DRV models showed the largest deviations from the observed diet, especially in their high beverage intakes. Often, optimization studies suggest major increases in beverages like water and tea, as these food groups provide nutrients without contributing to energy or fats and sugars. This issue can be dealt with by constraining the total quantity of all foods or beverages [25,57,62]. Excluding nutrient goals from allNUT-DRV in the modified lists (modNUT-DRV and modNUT-AR), especially for nutrients frequently enriched (e.g., salt with iodine) or poorly represented in the data (e.g., fluoride), improved this acceptability issue without a constraint on total quantity.

The RI covers the individual needs of 97,5% of the healthy population but also overestimates the needs of half the individuals. The AR, on the other hand, only meets the needs of 50% of a defined group of people, meaning that its population-wide application may increase the risk of deficiencies or malnutrition [63]. Regarding diet modeling, the advisory committee of the US Dietary Guidelines 2020 concluded that the RI is preferred for planning individual dietary recommendations and the AR is preferred at the population level, but needs to be adjusted appropriately and subsequently evaluated [64]. By contrast, in Australia, France, the Netherlands, and the Nordic Nutrition Recommendations, the RI was used to derive FBDGs [11–13,65,66].

Another strategy can be to apply the AR for specific nutrients only. With regards to iron, Nordman et al. compared the AR and RI and found that "initial optimizations indicated a difficulty in fulfilling the high iron recommendations of pre-menopausal women without imposing large changes in the diet [...]" [25]. To avoid compromising diet acceptability, a separate dietary pattern only for premenopausal women was calculated. This kind of target group segregation does not fit the goal of the German FBDG, which is supposed to include a dietary pattern applicable to the general adult population. Hence, the iron value for premenopausal women was used as the nutrient goal in the present model. In diet optimization studies for the Australian and Dutch FBDG, dietary intakes were modelled for the various target groups within the adult population and conflated a posteriori [11,13]. Furthermore, the share of plant-based foods has increased in these simple models; the potential of lower iron absorption in a more plant-based dietary pattern [67] is another argument in favor of using the higher target (RI), especially for this critical nutrient.

Overall, excluding nutrient goals not well represented in the data led to more realistic results and gave greater flexibility to the model. This allows to take other relevant aspects for FBDGs into account, such as diet–health relationships or environmental aspects [4]. However, a nutrient supply sufficient to meet the requirements of the population should be provided irrespective of these additional aspects. The modNUT-DRV approach seemed to provide the best compromise between the different requirements for nutrient goals in the German FBDG. The nutrients that were not regarded in the modified lists should still be considered in the final formulation of the FBDG, for example, recommendations for the use of salt fortified with iodine and fluoride.

## 4.1. Strengths and limitations

A variety of options are available to carry out diet modeling for FBDGs [68]; here, a systematic investigation of optimization options was undertaken. The main strength of the study is that it provides a rationale for decisions regarding the German FBDGs 2024 and could also guide other research using similar approaches. The study therefore adds to the body of data on the use of diet optimization for FBDGs. Although this simple model accounts only for nutrients

and acceptability, these are the most common components of diet optimization studies [8] and this framework can easily be expanded and applied to other use cases.

However, this study could only address some of the options in the context of diet modeling for FBDGs. With increasing model complexity, further decisions about the (mathematical) options need to be made, which were not addressed here. Whether such decisions challenge the current conclusions is unknown. In this work, we introduced various indicators for deviation from the observed diet. Data in the literature on this aspect is scarce and further investigation of the indicators is needed to confirm which ones give robust and meaningful results to compare various objective functions. Next, there are limitations of databases themselves, whether it is the age of the data (e.g., the NVS II was conducted 2005-2007) or their methodologies (e.g., under-reporting in dietary surveys, variations in nutrient compositions). Nonetheless, a diet optimization model may be easily updated whenever further or new data is available.

## 5. Conclusions

In this study, a novel diet optimization model was developed investigating crucial parameters such as the choice of decision variables, the mathematical form of the objective function, and nutrient goals. These insights will contribute to the framework not only for the German FBDG but also for decisions other FBDG-makers may face when using a diet modeling approach.

To answer the study objectives, only the nutritional and acceptability aspects for one target group of the German adult population were addressed. The integration of indicators for other dimensions of a sustainable diet, e.g., diet–health relations and environmental aspects, as a basis for the German FBDG 2024 will be part of a follow-up article.

## Supporting information

**S1 Table. Classification of whole grain and refined grain on all levels of FoodEx2.**
(XLSX)

**S2 Table. Working group of the German Nutrition Society for the development of the scientific basis for the German food-based dietary guideline 2024.**
(XLSX)

**S3 Table. All decision variables (food groups) on level 3 with according parent terms.**
(XLSX)

**S4 Table. Acceptability constraints on level 1.**
(XLSX)

**S5 Table. Share of population sub-groups used to calculate mean DRVs for adults aged 18-64.**
(XLSX)

**S6 Table. Optimization results for all analyses including food group quantities for the FBDG food groups, deviation indicators, and energy and nutrient contents.**
(XLSX)

## Acknowledgments

We thank Arno Lellmann for the thorough compilation of DRV values into a database, Theresa Maria Ting for her contribution to the calculation of the metrics for the deviation of dietary habits, and Julia Haardt for her contribution to the classification of whole grains food

groups in FoodEx2. We also thank Ute Alexy for her support with the LEBTAB data, Maike Benz for support in the dietary intake data analyses. We thank Katja Sandfuchs for her consent to publish the BLS-FoodEx2 matching as personal information. We thank Jan Raphael Schäfer for his support in compiling the database via SQL.

## Author contributions

**Conceptualization:** Anne Carolin Schäfer, Heiner Boeing, Kurt Gedrich, Anja Kroke, Jakob Linseisen, Stefan Lorkowski, Ute Nöthlings, Bernhard Watzl.

**Data curation:** Anne Carolin Schäfer, Rozenn Gazan, Johanna Conrad, Christina Breidenassel, Margrit Richter.

**Formal analysis:** Anne Carolin Schäfer.

**Methodology:** Anne Carolin Schäfer, Heiner Boeing, Rozenn Gazan, Florent Vieux, Bernhard Watzl.

**Software:** Rozenn Gazan, Anne Carolin Schäfer, Florent Vieux.

**Supervision:** Heiner Boeing, Johanna Conrad, Ute Nöthlings, Bernhard Watzl.

**Writing – original draft:** Anne Carolin Schäfer, Heiner Boeing.

**Writing – review & editing:** Anne Carolin Schäfer, Rozenn Gazan, Johanna Conrad, Kurt Gedrich, Christina Breidenassel, Hans Hauner, Anja Kroke, Stefan Lorkowski, Ute Nöthlings, Margrit Richter, Lukas Schwingshackl, Florent Vieux, Bernhard Watzl.

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
