## [Decision Letter · Decision Letter 0]

10 Dec 2024

PONE-D-24-43035A methodological framework for deriving the German food-based dietary guidelines 2024: food groups, nutrient goals, and objective functionsPLOS ONE

Dear Dr. Schäfer,

Thank you for submitting your manuscript to PLOS ONE. After careful consideration, we feel that it has merit but does not fully meet PLOS ONE’s publication criteria as it currently stands. Therefore, we invite you to submit a revised version of the manuscript that addresses the points raised during the review process.

We look forward to receiving your revised manuscript.

Kind regards,

Kuldeep Kumar Saxena

Academic Editor

PLOS ONE

Journal Requirements:

2. Please remove all personal information, ensure that the data shared are in accordance with participant consent, and re-upload a fully anonymized data set. 

Reviewers' comments:

Reviewer's Responses to Questions

**Comments to the Author**

1. Is the manuscript technically sound, and do the data support the conclusions?

Reviewer #1: Yes

2. Has the statistical analysis been performed appropriately and rigorously? 

Reviewer #1: Yes

3. Have the authors made all data underlying the findings in their manuscript fully available?

Reviewer #1: Yes

4. Is the manuscript presented in an intelligible fashion and written in standard English?

Reviewer #1: Yes

5. Review Comments to the Author

Reviewer #1: The author and the team studied on the "A methodological framework for deriving the German1 food-based dietary guidelines 2024: food groups, nutrient2 goals, and objective functions". The framework provides a comprehensive, structured approach to deriving dietary guidelines, balancing nutrition science, sustainability, and practicality. Addressing the concerns raised above, particularly around model validation, population-specific considerations, and sustainability trade-offs, will enhance the robustness and applicability of the guidelines. With some refinements, this framework has the potential to set a benchmark for national dietary guideline development in Europe. The research team has presented the work nicely in the introduction part however I would like to suggest to add some more references. The methodology part were presented nicely with proper food groups names and code. results obtained were discussed properly however its my suggestion to discuss results by comparing with the other models.

Overall author address the comments raised for final acceptance.

6. PLOS authors have the option to publish the peer review history of their article (what does this mean? ). If published, this will include your full peer review and any attached files.

**Do you want your identity to be public for this peer review?** For information about this choice, including consent withdrawal, please see our Privacy Policy .

Reviewer #1: **Yes: ** Dr. Devendra Kumar Pandey

---

## [Author Response · Author response to Decision Letter 1]

9 Jan 2025

Subject: Response to the editor's and reviewer's comments on the manuscript PONE-D-24-43035

Dear Prof. Dr. Kuldeep Kumar Saxena,

Dear Dr. Devendra Kumar Pandey,

Herewith we submit a revised version of our manuscript entitled "A methodological framework for deriving the German food-based dietary guidelines 2024: food groups, nutrient goals, and objective functions".

Thank you very much for your valuable review of our manuscript. We have thoroughly considered the editors' and reviewer's comments and revised our manuscript accordingly. Please find below a detailed point-by-point response to each of the comments and recommendations provided.

We hope to have met your expectations and thank you in advance for considering to publish the manuscript.

Yours sincerely,

Anne Carolin Schäfer, on behalf of all authors

Editor's comments

1. "Please ensure that your manuscript meets PLOS ONE's style requirements, including those for file naming."

Thank you for the notice. We changed the following points:

- Formatting changes: No bold letters in article title and manuscript title centered.

- Line 104: Fig 1 instead of Figure 1, and the file was renamed to "Fig1.tif"

- Line 110: Made the figure caption bold instead of italic and adapted the figure caption.

- Line 192: Fig 1 instead of Figure 1.

- Line 320: Deleted "see" before "Table 3"

- Level 3 headings were changed to 14 pt font.

- Supporting file information in the end (lines 861-870) was changed to bold letters.

- Supporting information file was renamed to "S1-S6_Table".

- Author contributions and short title were removed from the "Manuscript" file.

2. "Please remove all personal information, ensure that the data shared are in accordance with participant consent, and re-upload a fully anonymized data set."

Thank you for pointing this out. We removed the personal information from lines 187-188 and acknowledged the colleague's work in the acknowledgements instead (lines 621-622).

3. "Please note that PLOS ONE has specific guidelines on code sharing for submissions in which author-generated code underpins the findings in the manuscript. In these cases, we expect all author-generated code to be made available without restrictions upon publication of the work. Please review our guidelines at https://journals.plos.org/plosone/s/materials-and-software-sharing#loc-sharing-code and ensure that your code is shared in a way that follows best practice and facilitates reproducibility and reuse."

and

4. "We note that you have indicated that there are restrictions to data sharing for this study. For studies involving human research participant data or other sensitive data, we encourage authors to share de-identified or anonymized data. However, when data cannot be publicly shared for ethical reasons, we allow authors to make their data sets available upon request. For information on unacceptable data access restrictions, please see http://journals.plos.org/plosone/s/data-availability#loc-unacceptable-data-access-restrictions.

Please update your Data Availability statement in the submission form accordingly."

Thank you for pointing this out. We share the position that data being used to produce the article should be in principle publicly available. However, in our case, we did not generate data in the context of this article that were not shown in the tables and supplements. The main data sources were from third parties that can easily be contacted via the information given in the text and references. The main data sources that have been used are listed below:

Software:

• DGE, parts of own developed program solutions can be made available upon request (contact at https://www.dge.de/)

• MS-Nutrition has provided the basic programming and should be consulted if interested in the program code (contact at https://ms-nutrition.com/en/)

Data:

• Nutrient database: The data on nutrients used for obtaining the results presented in the manuscript are available for the Bundeslebensmittelschlüssel (BLS) from the Max Rubner-Institut (contact: https://blsdb.de/contact) and for LEBTAB from the University of Bonn (https://www.epi.uni-bonn.de/forschung/donald-studie, contact via noethlings@uni-bonn.de).

• Food intake data: The data on food intake underlying the results presented in the study are available from the European Food Safety Authority (EFSA) which has published national consumption data at https://www.efsa.europa.eu/en/data-report/food-consumption-data and can be accessed via a public access per data and documents request (for details, see https://www.efsa.europa.eu/sites/default/files/2023-08/pad-guidance-for-applicants.pdf)

All relevant data on changes applied to the original databases (e.g. whole grain classification) are mentioned within the manuscript and its supporting information files.

5. "Please review your reference list to ensure that it is complete and correct. If you have cited papers that have been retracted, please include the rationale for doing so in the manuscript text, or remove these references and replace them with relevant current references. Any changes to the reference list should be mentioned in the rebuttal letter that accompanies your revised manuscript."

We checked the availability of all internet links and updated the date of citation accordingly for references:

• Australian Government, National Health and Medical Research Council. A modelling system to inform the revision of the Australian guide to healthy eating. 2011

• EFSA (European Food Safety Authority). The food classification and description system FoodEx2 (revision 2). 2015

• EFSA (European Food Safety Authority). Overview on tolerable upper intake levels as derived by the Scientific Committee on Food (SCF) and the EFSA Panel on Dietetic Products, Nutrition and Allergies (NDA). 2018

• R Core Team. R: A language and environment for statistical computing. Vienna, Austria: 2022

• Berkelaar M. lpSolve: Interface to ‘Lp_solve’ v. 5.5 to Solve Linear/Integer Programs. 2023

• Berwin A, Turlach R, Weingessel A. quadprog: functions to solve quadratic programming problems. Comprehensive R Archive Network (CRAN) 2019

• FAO (Food and Agriculture Organization), WHO (World Health Organization). FAO/WHO Global Individual Food consumption data Tool (FAO/WHO GIFT). Project brief updated as per August 10th, 2017.

• Ministry of Agriculture Zambia. Zambia Food-Based Dietary Guidelines. Technical recommendations. 2021

• Rubens K, Neven L, Jonckheere J. Food and environmentally responsible consumption: towards healthy food patterns for a healthy planet. Background document for the food triangle recommendations. 2021

• FAO (Food and Agriculture Organization of the United Nations). Food-based dietary guidelines. Third webinar: Diet modelling for food-based dietary guidelines. 2019

We replaced "Tufts University. GDD 2018 Dietary Data Harmonization. Global Dietary Database. 2019" by " Karageorgou D, Lara Castor L, Padula de Quadros V, et al. Harmonising dietary datasets for global surveillance: methods and findings from the Global Dietary Database. Public Health Nutrition. 2024" which is the scientific publication for this study.

6. "While revising your submission, please upload your figure files to the Preflight Analysis and Conversion Engine (PACE) digital diagnostic tool, https://pacev2.apexcovantage.com/. PACE helps ensure that figures meet PLOS requirements."

Thank you for raising this issue. We uploaded Figure 1 in PACE and changed the format accordingly: The resolution was changed to 300 PPI and the file was converted to a valid TIF file.

Reviewers' comments

1. "The author and the team studied on the "A methodological framework for deriving the German1 food-based dietary guidelines 2024: food groups, nutrient2 goals, and objective functions". The framework provides a comprehensive, structured approach to deriving dietary guidelines, balancing nutrition science, sustainability, and practicality. Addressing the concerns raised above, particularly around model validation, population-specific considerations, and sustainability trade-offs, will enhance the robustness and applicability of the guidelines. With some refinements, this framework has the potential to set a benchmark for national dietary guideline development in Europe. The research team has presented the work nicely in the introduction part however I would like to suggest to add some more references."

Thank you for your careful assessment of the paper and raising important issues. We revised the introduction thoroughly and added the following references:

• Line 60: WHO 1998: Preparation and use of food-based dietary guidelines.

• Line 61: WHO 2003: Food-based dietary guidelines in the WHO European Region.

• Line 62: Gonzalez Fischer et al. 2016: Plates, pyramids, and planets. Developments in national healthy and sustainable dietary guidelines: a state of play assessment.

• Line 66: Schäfer et al. 2020: Integration of various dimensions in food-based dietary guidelines via mathematical approaches.

Wilson et al. 2019: Achieving healthy and sustainable diets: a review of the results of recent mathematical optimization studies

• Line 84: Mariotti et al. 2021 Perspective: modeling healthy eating patterns for food-based dietary guidelines-scientific concepts, methodological processes, limitations, and lessons.

Nordman et al. 2023: Exploring healthy and climate-friendly diets for Danish adults: an optimization study using quadratic programming

2. "The methodology part were presented nicely with proper food groups names and code. results obtained were discussed properly however its my suggestion to discuss results by comparing with the other models.

Overall author address the comments raised for final acceptance."

We thank the reviewer for these valuable remarks. We agree with the reviewer that the other models for deriving food-based dietary guidelines (FBDGs) that were mentioned in the introduction should also be discussed in more detail. However, our study addresses methodological aspects of singular parameters of the final model, not the final model for the German FBDG itself whilst the other countries' publications on their FBDG optimization models were mostly information on final modelling decisions and the final FBDGs. For comparing our results on basic methodological questions, such as the choice of acceptability constraints, we have mostly chosen publications with results on similar research questions, such as Borgonjen et al.

Nonetheless, in the beginning of the discussion (lines 382-393), we have added information on the differences in the approaches for diet modelling for FBDGs as follows:

• "This study provides methodological investigations of specific aspects of diet optimization which we were faced with in the process to tailor the final model for the German FBDG. We could not find similar methodological comparisons of different settings when studying other country's FBDGs. This could be due to the fact that in most other FBDGs based on optimization results such as the FBDGs in France, Australia, and for Malawian children, iterative approaches to introduce new parameters to the models were used instead of comparisons of different parameters settings [11,12,14]. We found the mentioning of sensitivity analyses for different objective functions in terms of minimizing deviation from the current diet or from certain nutrients were carried out, but results were not reported [14] or the food groups were studied in preliminary optimization testing, but again no methodological details or insights of this exercise were given [11]. For the Dutch FBDG, only the final selected settings were reported without comparisons with alternative scenarios [13]."

and have included results from the other countries' publications where applicable as follows:

• Lines 403-404: We added characteristics for the selection of food groups that were mentioned in the FBDG optimization studies: "…, determined considering characteristics such as nutrient composition or dietary habits".

• Lines 413-415: "To the best of our knowledge, a hierarchical system like FoodEx2 has not been applied to diet optimization, neither systematically studied as decision variables."

• Line 446: "diet modelling for the Australian FBDG"

• Lines 467-468: We added the reference for a recently published conference abstract on a review of acceptability in diet optimization studies and the decisions from the aforementioned national FBDG: "and in the French, Australian, and Malawian FBDG for children [11,12,14],"

• Line 469: We specified that the cited study is the one for the Dutch FBDG.

• Line 494: We specified "mathematical" type of objective function

• Lines 553: We added the decision for the Australian FBDG: "By contrast, in Australia, France, the Netherlands, and the Nordic Nutrition Recommendations, the RI was used to derive FBDGs [11,13,64].". We also added another citation for the choice of the French DRVs: French Agency for Food, Environmental and Occupational Health & Safety (ANSES), editor. Updating of the PNNS guidelines: revision of the food-based dietary guidelines. (Line 555).

• Lines 563-565: We added information on how other countries dealt with different nutrient goals for different population groups: "In diet optimization studies for the Australian and Dutch FBDG, dietary intakes were modelled for the various target groups within the adult population and conflated a posteriori [11,13]."

---

## [Editor Report · Decision Letter 1]

14 Jan 2025

A methodological framework for deriving the German food-based dietary guidelines 2024: food groups, nutrient goals, and objective functions

PONE-D-24-43035R1

Dear Dr. Schäfer,

We’re pleased to inform you that your manuscript has been judged scientifically suitable for publication and will be formally accepted for publication once it meets all outstanding technical requirements.

Kind regards,

Kuldeep Kumar Saxena

Academic Editor

PLOS ONE
---

## [Editor Report · Acceptance letter]

PONE-D-24-43035R1

PLOS ONE

Dear Dr. Schäfer,

I'm pleased to inform you that your manuscript has been deemed suitable for publication in PLOS ONE. Congratulations! Your manuscript is now being handed over to our production team.

Kind regards,

on behalf of

Dr. Kuldeep Kumar Saxena

Academic Editor

PLOS ONE